# Synthesis of Bioactive Silver Nanoparticles by a *Pseudomonas* Strain Associated with the Antarctic Psychrophilic Protozoon *Euplotes focardii*

**DOI:** 10.3390/md18010038

**Published:** 2020-01-03

**Authors:** Maria Sindhura John, Joseph Amruthraj Nagoth, Kesava Priyan Ramasamy, Alessio Mancini, Gabriele Giuli, Antonino Natalello, Patrizia Ballarini, Cristina Miceli, Sandra Pucciarelli

**Affiliations:** 1School of Biosciences and Veterinary Medicine, University of Camerino, 62032 Camerino, Italy; sindhuramaria@gmail.com (M.S.J.); amruthjon@gmail.com (J.A.N.); kesavanlife@gmail.com (K.P.R.); alessio.mancini@unicam.it (A.M.); patrizia.ballarini@unicam.it (P.B.); cristina.miceli@unicam.it (C.M.); 2School of Sciences and Technology, University of Camerino, 62032 Camerino, Italy; gabriele.giuli@unicam.it; 3Department of Biotechnology and Biosciences, University of Milano-Bicocca, 20126 Milano, Italy; antonino.natalello@unimib.it

**Keywords:** green synthesis biomaterials, silver nitrate, antibiotics, nanotechnology

## Abstract

The synthesis of silver nanoparticles (AgNPs) by microorganisms recently gained a greater interest due to its potential to produce them in various sizes and morphologies. In this study, for AgNP biosynthesis, we used a new *Pseudomonas* strain isolated from a consortium associated with the Antarctic marine ciliate *Euplotes focardii*. After incubation of *Pseudomonas* cultures with 1 mM of AgNO_3_ at 22 °C, we obtained AgNPs within 24 h. Scanning electron (SEM) and transmission electron microscopy (TEM) revealed spherical polydispersed AgNPs in the size range of 20–70 nm. The average size was approximately 50 nm. Energy dispersive X-ray spectroscopy (EDS) showed the presence of a high intensity absorption peak at 3 keV, a distinctive property of nanocrystalline silver products. Fourier transform infrared (FTIR) spectroscopy found the presence of a high amount of AgNP-stabilizing proteins and other secondary metabolites. X-ray diffraction (XRD) revealed a face-centred cubic (fcc) diffraction spectrum with a crystalline nature. A comparative study between the chemically synthesized and *Pseudomonas* AgNPs revealed a higher antibacterial activity of the latter against common nosocomial pathogen microorganisms, including *Escherichia coli, Staphylococcus aureus* and *Candida albicans*. This study reports an efficient, rapid synthesis of stable AgNPs by a new *Pseudomonas* strain with high antimicrobial activity.

## 1. Introduction

Nanotechnology has become an emerging field in the area of biotechnology, dealing with the synthesis, design and manipulation of particles with approximate sizes from 1 to 100 nm. Nanoparticles (NPs) are used in biomedical sciences, healthcare, drug–gene delivery, space industries, cosmetics, chemical industries, optoelectronics, etc. [1]. Various physiochemical methods have been used for AgNP synthesis, including microwave, biochemical and electrochemical synthesis, chemical reduction (aqueous and non-aqueous), irradiation, ultrasonic-associated, photo-induced, photo-catalytic and microemulsion methods. However, these methodologies have various disadvantages because they imply high energy consumption and the use of toxic reagents with the generation of hazardous waste, which causes potential risks to the environment and human health [2,3]. In these days, there is a growing need to develop simple, cost-effective, reliable, bio-compatible and eco-friendly approaches for the synthesis of nanomaterials, which do not contain toxic chemicals in the synthesis protocols. For mining the metallic nanomaterials, microbial-mediated green synthesis has recently been considered as a promising source [4]. Green synthesis of nanoparticles represents a cost-effective and environmentally friendly method with advantages over conventional methods that involve chemical, potentially toxic solvents. For green NP synthesis, the most important issues are the solvent medium combined with the selection of ecologically nontoxic, reducing and stabilizing agents [5]. There are different green methods for nanoparticle synthesis, but the most commonly appreciated is through bacteria, because bacteria are usually easy to grow [6]. Capping agents are considered fundamental for nanoparticles stabilization. Capped AgNPs are known to exhibit better antibacterial activity with respect to uncapped AgNPs [7]. Biologically synthesized nanoparticles have remarkable potential since they can be easily coated with a lipid/protein layer, which confers physiological solubility and stability useful for applications in biomedicine [8]. 

AgNPs possess general antibacterial and bactericidal properties. These are mostly against methicillin-resistant strains [9]. Gram-negative and Gram-positive bacteria are relevant causes of numerous infections in hospitals. Due to the increased microbial resistance to multiple antibiotics [10], many researchers are interested in developing novel and effective antimicrobial agents [11]. Furthermore, AgNPs exhibit anti-biofilm activities [12] and synergistic activities with diverse antibiotics, such as β-lactams, macrolides and lincosamides [13]. The use of silver-coated antiseptics shows a broad-spectrum activity and a far lower chance than antibiotics in inducing the typical microbial resistance [14].

The mechanism at the basis of the extracellular synthesis of nanoparticles using microbes appears based on enzymes such as the secreted nitrate reductase that helps in the production of metal nanoparticles from metal ions [15]. Such mechanism was shown in *Bacillus licheniformis* [16]. The biosynthesis and stabilization of nanoparticles in *Stenotrophomonas maltophilia* via charge capping, involving the electron shuttle enzymatic metal reduction process produced by the Nicotinamide Adenine Dinucleotide Phosphate (NADPH)-dependent reductase enzyme, has also been reported [17]. In *Pseudomonas* spp.*,* a possible process for the biosynthesis of AgNPs is described, to be performed by a Nicotinamide Adenine Dinucleotide (NADH)-dependent nitrate reductase [18,19]. The enzyme may be responsible for the reduction of Ag^+^ to Ag^0^ and the subsequent formation of AgNPs, where the NADH-dependent reductase is expected to act as a carrier while the bioreduction occurs by means of the electrons from NADH [18,19].

The aim of this study is to develop a simple and low-cost approach for AgNP intracellular synthesis using a new Pseudomonas strain isolated from a consortium associated with the psychrophilic marine ciliated protozoon Euplotes focardii [20], which we named Pseudomonas sp. ef1 [21]. E. focardii is a free-swimming ciliate endemic of the oligotrophic coastal sediments of the Antarctic Terra Nova Bay [22]. It has been maintained in cultures for more than 20 years after the first isolation. E. focardii’s optimal growing temperature is about 4–5 °C, with a drop at 8–10 °C and not surviving if exposed to temperatures over 10 °C [22]. We showed that the AgNPs produced by Pseudomonas sp. ef1 possess higher antimicrobial activity with respect to those chemically synthesized, which could be used against common pathogenic microorganisms. 

## 2. Results and Discussion

### 2.1. Biosynthesis of AgNPs and UV–Vis Spectroscopy Characterization

We primarily investigated the biosynthesis of AgNPs by *Pseudomonas* sp. ef1 through the observation of the culture medium colour change by incubation of the bacterial biomass with 1 mM AgNO_3_ at 22 °C (the optimal growing temperature of *Pseudomonas* sp. ef1). A change in colour from white to brown occurred within 24 h in the presence of light (Appendix A). The change to a brown colour of the culture was maintained for 72 h (Appendix A). No colour change was observed in the control culture containing the heat-killed bacterial biomass with 1 mM AgNO_3_ (data not shown). 

Medium colour change related to extracellular synthesis of AgNPs has been previously reported in another *Pseudomonas* culture [23], as well as in *Bacillus methylotrophicus* [24] and *Actinobacteria* SL19 and SL24 strains, and in fungi as Fusarium semitectum, *Aspergillus fumigatus* [25] and Streptomyces sp. The culture medium colour change is attributed to the excitation of the surface plasmon resonance of AgNPs [26,27]. 

*Pseudomonas* sp. ef1 AgNPs formation was confirmed by UV–vis spectroscopy, considered one of the most valuable methods for the characterization of the optical response of metal nanoparticles, including AgNPs. This method has been demonstrated to be appropriately sensitive to check AgNPs’ intense surface plasmon resonances (SPRs) [28] in the range of 350–600 nm [29,30,31]. A 0.1 mL aliquot of the *Pseudomonas* sp. ef1 culture was diluted with 0.9 mL of ddH2O and UV–visible spectra was recorded from 300 to 800 nm wavelength at room temperature: A relevant peak at about 420 nm was found (Appendix A), as also reported for AgNPs produced by *Pseudomonas* putida NCIM 2650 [32] and Pseudomonas sp. (JQ989348) [33]. By contrast, *Pseudomonas* sp. “ram bt-1” AgNPs showed absorbance spectra at 430 nm [34]. The presence of a single SPR peak suggests AgNPs of spherical shape [35]. 

### 2.2. Morphology and Chemical Composition of Pseudomonas sp. ef1 AgNPs 

We performed scanning electron microscopy (SEM) to define the size and shape of the AgNPs synthesized by *Pseudomonas* sp. ef1. SEM images (Appendix A) revealed polydispersed (i.e., non-uniform in size) AgNPs of spherical shape. Their size ranged from 20 to about 100 nm.

To better understand the surface morphology and for getting additional information on the size, a TEM investigation was conducted (Figure 1). Aliquots of AgNP solution were placed onto a nitrocellulose- and Formvar-coated copper grid and maintained to dry under room conditions. TEM micrographs suggested particle sizes around 10 nm (clearly visible in Figure 1A) to 70 nm (Figure 1B), with the average size being 50 nm. The particles showed a spherical shape, well separated from each other even when these formed aggregates, suggesting the presence of capping peptides around each particle, whose role is to stabilize the nanoparticles.

*Pseudomonas* sp. ef1 AgNPs appear similar to those produced by other *Pseudomonas* [36,37,38,39,40,41,42]. The TEM grid analysis of *Pseudomonas* sp. ef1 bio-AgNPs revealed smooth-surfaced polydispersed particles, approximately spherical in shape with the size ranging from 12.5 to 100 nm. By contrast, bio-AgNPs from Pseudomonas putida were monodispersed and smaller in size (6 to 16 nm) [43]. 

The chemical composition of the *Pseudomonas* sp. ef1 AgNPs was obtained by energy dispersive X-ray (EDX) spectrum analysis (Figure 2). We observed an intense signal of Ag at 3 keV, which confirmed the presence of AgNPs. Metallic AgNPs are typically reported to show a strong signal peak at 3 keV, due to surface plasmon resonance [44,45]. However, other element (C, N and O) signals were detected at normal mode (Figure 2 and Table 1). These elements probably derive from the emissions of the capping proteins. The EDX spectrum analysis of Pseudomonas fluorescens CA 417 AgNPS also showed the presence of a high intensity absorption peak at 3 keV [46]. 

### 2.3. Capping Proteins and Crystalline Structure of Pseudomonas sp. AgNPs

To confirm the potential interactions between the silver salts and the capping proteins, which could account for the reduction of Ag^+^ ions with consequent stabilization of AgNPs, we performed FTIR measurements (Figure 3A)**.** The amide linkages between the amino acid residues in proteins produce a typical signature in the infrared spectral region [47].

The FTIR spectrum of *Pseudomonas* sp. ef1 AgNPs is characterized by the protein Amide A, B, I, II and III bands (Figure 3). In particular, the peaks around 3280 cm^−1^ (Amide A) and 3070 cm^−1^ (Amide B) are mainly assigned to the NH vibrations. The absorption maximum of the Amide I band, due to the C=O stretching of the peptide bond, occurs around 1632 cm^−1^. The Amide II band, mainly due to the amide NH bending, peaked around 1538 cm^−1^. The complex absorption in the 1200–950 cm^−1^ spectral region can be tentatively assigned to carbohydrate absorption. The 1740 cm^−1^ peak is characteristic of C=O carbonyl groups [48].

The overall FTIR pattern confirms that capping proteins are present in the AgNPs and that these proteins are not extensively aggregated. Protein–nanoparticle interactions are produced either through free amine groups or cysteine residues and through the electrostatic attraction of negatively charged carboxylate groups, specifically in enzymes [49]. The free amine and carbonyl groups of bacterial proteins could possibly be responsible for the formation and stabilization of AgNPs [50,51]. 

The capping proteins prevent aggregation and provide nanoparticle stability [52]. Fourier transform infrared spectrum applied to the deep-sea bacterium Pseudomonas sp. JQ989348 AgNPs showed the presence of proteins in large amounts, as well as other secondary metabolites [33]. 

The crystalline structure of Pseudomonas sp. ef1 AgNPs was confirmed by XRD analysis (Figure 3B). The AgNPs diffraction spectrum showed a face-centred cubic (fcc) crystalline nature, including peaks at 38.95°, 45.12°, 65.39° and 78.12° (labelled as Ag in Figure 3B), which corresponded to plane values of (1 1 1), (2 0 0), (2 2 0) and (3 1 1) at the 2θ angle. These were consistent with the standard data JCPDS file no. 01-087-0717. Peaks indicated by asterisks in Figure 3B probably correspond to the crystallization of the bio-organic phase occurring on the AgNPs surface, as also reported in [53,54,55,56,57]. Alternatively, these peaks may be due to AgNO3, which has not been reduced by Pseudomonas sp. ef1. 

X-ray diffraction (XRD) analysis of *Pseudomonas* fluorescens CA 417 AgNPs also revealed well-defined peaks at 38° 44°, 64° and 78°, thus showing the face-centred cubic (fcc) metallic crystal corresponding to the (111), (200), (220) and (311) facets of the crystal planes at the 2θ angle [46]. The crystalline structure of biogenic AgNPs of Pseudomonas putida MVP2 was also confirmed by XRD [43].

### 2.4. Pseudomonas sp. ef1 AgNPs Antimicrobial Activity 

The antibacterial activity of *Pseudomonas* sp. ef1 AgNPs was tested against pathogenic Gram-positive and Gram-negative bacteria, as well as fungi, and compared with that of chemically synthesized AgNPs and AgNO_3_ (Appendix A and Figure 4). In the comparative study, the biosynthesized AgNPs exhibited a stronger antibacterial property than the chemically synthesized AgNPs and AgNO_3_ (Figure 4). Among Gram-negative bacteria, the larger inhibition zone (Ø 19.0 mm) was against *Escherichia coli* and the smallest (Ø 14.0 mm) was against *Pseudomonas aeruginosa* and *Serratia marcescens*. Among Gram-positive bacteria, the highest zone of 15 mm was formed against *Staphylococcus aureus*. Among fungi, the larger zone was against *Candida albicans* (Ø 15.0 mm).

It has been reported that *E. coli* showed a greater sensitivity by comparison with that of *Bacillus cereus* and *Streptococcus pyogene*, probably due to the narrow cell walls of Gram-negative bacteria with respect to Gram-positive bacteria [58].

The use of the biosynthesized AgNPs may be one of the promising approaches to overcome bacterial resistance and could also play a new key role in pharmacotherapeutics. The mechanism of the AgNP-mediated bactericidal property is still to be understood. A mechanism proposed by other studies is that AgNPs attach to the cell wall, thus modifying the membrane integrity and disturbing its permeability and cell respiration functions [59,60]. Most likely, the antibacterial activity of AgNPs is size dependent. This means that smaller AgNPs that have a large surface area available for interactions function as more efficient antimicrobial agents than larger ones. It is also possible that AgNPs can penetrate inside the bacteria and not only interact with the membrane of the cell [60]. Another possible process responsible of AgNPs antimicrobial activity may be the release of Ag^+^ ions, since they may play a partial but relevant role in the bactericidal effect [60].

## 3. Materials and Methods 

### 3.1. Synthesis of AgNPs by Pseudomonas sp. ef1

The bacterial biomass was produced by inoculating *Pseudomonas* sp. ef1 into Luria-Bertani (LB) medium (10 g of Tryptone, 10 g of NaCl and 5 g of yeast extract, dissolved in 1 L of ddH_2_0). The culture flasks were incubated on an orbital shaker set at 220 rpm, at 22 °C. After 24 h the biomass was harvested by centrifuging at 5000 rpm for 30 min. After removal of the supernatant, approximately 2 mg of the bacterial biomass was transferred into an Erlenmeyer flask containing a solution of 1 mM AgNO_3_. The mixture was placed in the orbital shaker set at 200 rpm for 24 h at 22 °C. The heat-killed biomasses incubated with silver nitrate were maintained as control. Biosynthesis was carried out in bright condition as visible light irradiation is known to increase the biosynthetic rate of AgNPs formation. The bioreduction of Ag^+^ ions were monitored by changes in colour of the bacterial biomass reaction mixture containing the AgNO_3_ (Appendix A) and by UV–visible spectroscopy (UV-1800, Shimadzu): a 0.1 mL aliquot of the sample was diluted with 0.9 mL of ddH_2_O and UV–visible spectra was recorded from 300 to 800 nm wavelength. ddH_2_O was used as blank.

### 3.2. Purification of AgNPs

Bacterial biomass was collected by centrifugation at 5000 rpm for 30 min. The resulting pellet was then suspended in ddH_2_O and ultra-sonicated at a pulse rate of 6V at intervals of 30 s for ten cycles. Afterwards, the solution was centrifuged again at 5000 rpm for 30 min and the supernatant loaded on a Sephadex G-50 resin equilibrated in 10 mM Tris buffer (pH 7.0) to remove contaminating debris and proteins. AgNPs were finally extracted from the buffered solution by adding 3 volumes of isopropanol to the obtained nanoparticle solution. Isopropyl alcohol is known to dissolve a wide range of non-polar compounds and to evaporate quickly compared to ethanol. The mixture was rotated on the orbital shaker overnight and subjected to evaporation to obtain a purified powdered highly enriched in NPs.

### 3.3. Chemical Synthesis of AgNPs 

A solution of 1 mM of AgNO_3_ was heated to boil. As the solution started to boil, a sodium citrate solution was added drop-by-drop until the solution turned into a greyish-yellow color, indicating Ag^+^ ion formation. Heating was continued for 60 s. The solution was then chilled to room temperature.

### 3.4. Scanning Electron Microscopy (SEM), Transmission Electron Microscopy (TEM) and Energy Dispersive X-ray Analysis (EDAX)

For SEM (ZIESSA, Sigma 300) analysis, purified AgNPs were sonicated for 15 min to reach a uniform distribution. A drop of the solution was loaded on carbon-coated copper grids and allowed to evaporate under infrared light for 30 min. 

TEM (PHILIPS EM208S) analysis was performed using an acceleration voltage of 100 kV. Drops of an AgNP solution were loaded on nitrocellulose- and Formvar-coated copper TEM grids. After 2 min, the extra solution was removed, and the grids were allowed to dry at room temperature. The acquired data were analysed by Statistical Software (StatSoft, Tulsa, Okla., United States) using the variability plot of average methods. After 100 measurements the size distribution of the AgNPs was estimated using TEM imaging and analysis software (TIA).

EDAX analysis of AgNPs was performed using Field Emission Scanning Electron Microscope (FESEM) equipped with an EDAX attachment.

### 3.5. Fourier Transform Infrared Spectroscopy (FTIR) Analysis

AgNPs were deposed on the single reflection diamond element of the attenuated total reflection (ATR) device (Quest, Specac) and dried at room temperature. The ATR/FTIR spectrum was collected by the Varian 670-IR spectrometer, equipped with a nitrogen-cooled Mercury Cadmium Telluride detector, under the following conditions: triangular apodization, scan speed of 25 kHz, resolution of 2 cm^−1^ and 512 scan co-additions [61]. 

### 3.6. X-ray Diffraction Analysis (XRD)

X-ray Diffraction measurements were performed by scanning drop-coated films of AgNps in a wide range of Bragg angle 2θ at a rate of 2 min^–1^. A Philips PW 1830 instrument was used, and it was operated at a voltage of 40 kV with a current of 30 mA using monochromatic Cu Kα radiation (λ = 1.5405 Å). The diffracted intensities were recorded in the 2θ range of 10°–80°. To elucidate the crystalline structure, the resulting images were compared with the Joint Committee on Powder Diffraction Standards (JCPDS) library. 

### 3.7. Screening of Antimicrobial Activity of Biosynthesized AgNps

The evaluation of the biosynthesized AgNPs antibacterial activity was carried out by the Kirby–Bauer disc diffusion method on the following twelve stains: Staphylococcus aureus, Staphylococcus epidermidis, Streptococcus agalactiae (Gram-positive bacteria); *Escherichia coli, Klebsiella pneumonia, Pseudomonas* sp*., Proteus mirabilis, Citrobacter koseri, Acinetobacter baumanii, Serratia marcescens* (Gram-negative bacteria); and *Candida albicans* and *Candida parapsilosis* (fungi). 

The biosynthesized AgNPs were tested for antibacterial activity by the Kirby–Bauer disc diffusion method. The pathogenic cultures were subcultured into peptone broth and incubated at 37 °C to reach 10^5^–10^6^ CFU ml^−1^. The fresh cultures of pathogens were plated using a sterile cotton swab on Petri dishes containing Muller Hinton Agar. Six millimetre filter paper disks impregnated with 25 μL of biosynthesized AgNPs using a sterile micropipette were placed on the pathogen-plated agar. Bio-AgNPs disks were compared with chemically synthesized AgNPs disks. AgNO_3_ disks and distilled water disks were used as control. The plates were incubated at 37 °C for 18–24 h to measure the zone of inhibition.

## 4. Conclusions

In this study we reported an easy and efficient biological method to synthesize AgNPs using the biomass of a novel *Pseudomonas* strain isolated from a bacterial consortium found in association with the Antarctic ciliate *E. focardii*. We also characterized these AgNPs. Their stability makes the present method a viable alternative to chemical synthesis methods. Due to the lesser specificity of the reaction parameters, this process can be explored for large-scale synthesis of AgNPs. The study highlights an efficient strategy to obtain bionanomaterial that can be used against a large number of drug resistant pathogenic bacteria, thus contributing to solve this globally serious concern, especially given there being a limited choice of antibiotic treatment [62]. Furthermore, these AgNPs show the highest antimicrobial activity with respect to those that are chemically synthesized. The *Pseudomonas* strain here used can also be exploited to remove silver nitrate contamination from the environment, allowing to associate its potential in bioremediation and in antibiotics production.

## 5. Patents

The results of this paper are related to the patent number 102019000014121 deposited in 06/08/2019.

## Figures and Tables

**Figure 1 marinedrugs-18-00038-f001:**
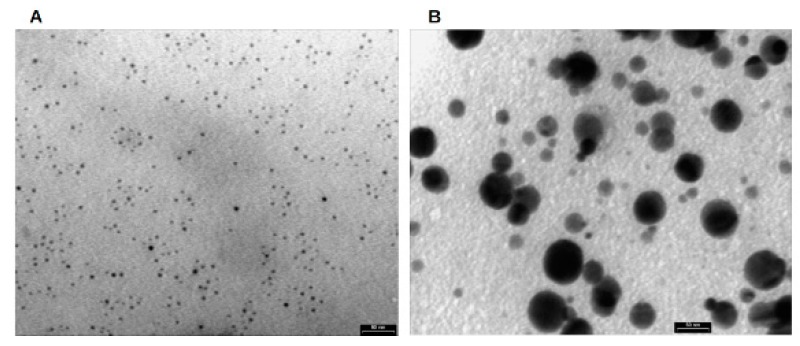
Transmission electron microscopy (TEM) micrographs of *Pseudomonas* sp. ef1 silver nanoparticles (AgNPs). The particles show a spherical shape with size from 10 nm (**A**) to 70 nm (**B**), with the average size being 50 nm. Bars: 50 nm

**Figure 2 marinedrugs-18-00038-f002:**
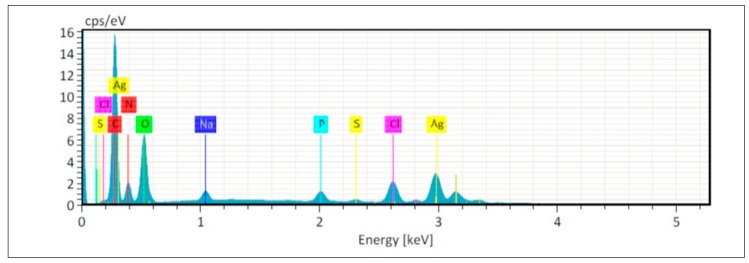
EDAX investigation of *Pseudomonas* sp. ef1 AgNPs. A: EDAX spectrum of AgNPs; Ag, C, N and O indicate the silver (the highest peak, recorded at 3 keV), carbon, nitrogen and oxygen signals (the relative amounts are reported in Table 1).

**Figure 3 marinedrugs-18-00038-f003:**
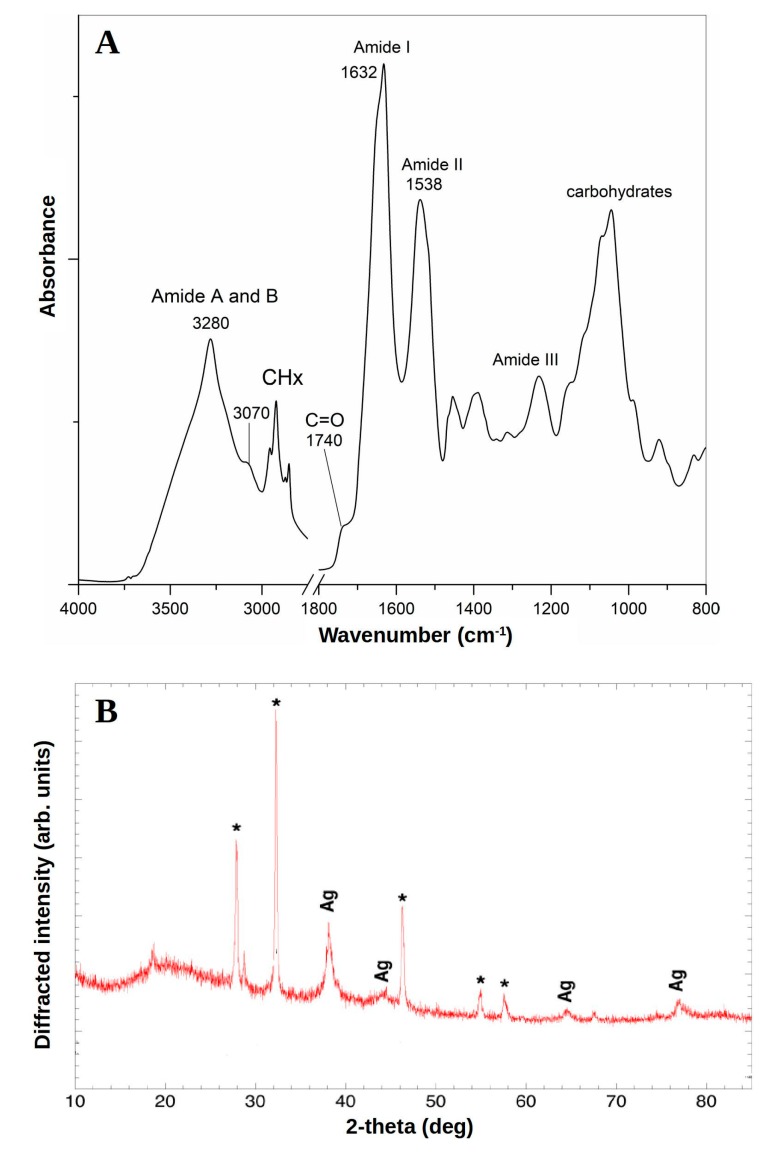
(**A**) FTIR absorption spectrum of *Pseudomonas* sp. ef1 AgNPs. The peak position and the assignment of the main components are reported. (**B**) XRD spectrum recorded for *Pseudomonas* sp. ef1 AgNPs. Four intense peaks at 38.95°, 45.12°, 65.39° and 78.12° correspond to plane values of (1 1 1), (2 0 0), (2 2 0) and (3 1 1) at the 2θ angle, which were consistent with the standard data JCPDS file no. 01-087-0717, indicated by asterisks.

**Figure 4 marinedrugs-18-00038-f004:**
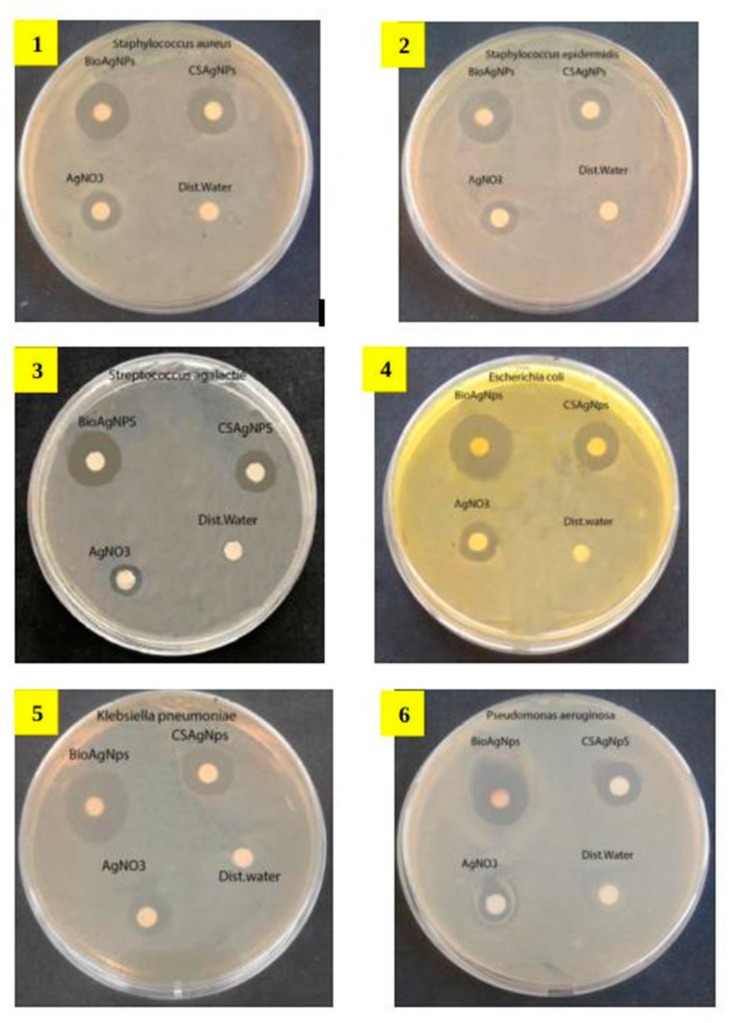
Antimicrobial activity of *Pseudomonas* sp. ef1 AgNPs tested against twelve human pathogens. The bio AgNPs disks were compared with chemically synthesized AgNPs disks. AgNO_3_ disks and distilled water disks were used as control. The human pathogens are: 1. *Staphylococcus aureus*, 2. *Staphylococcus epidermidis*, 3. S*treptococcus agalactie*, 4. *Escherichia coli,* 5. *Klebsiella pneumonia*, 6. *Pseudomonas aeruginosa*, 7. *Proteus mirabilis*, 8. *Citrobacter koseri*, 9. *Acinetobacter baumanii*, 10. *Serratia marcescens*, 11. *Candida albicans*, 12. *Candida parapsilosis*.

**Table 1 marinedrugs-18-00038-t001:** Quantitative EDAX results of *Pseudomonas* sp. ef1 AgNPs.

Element	At.No	Netto	Mass[%]	Mass Norm[%]	Atom[%]	Abs. err [%]1 sigma	rel err [%]1 sigma
Carbon	6	197121	22.22	29.78	48.86	2.45	11.01
Nitrogen	7	27658	7.08	9.49	13.35	0.90	12.76
Oxygen	8	90933	15.84	21.22	26.15	1.82	11.48
Sodium	11	16435	1.20	1.61	1.38	0.10	8.09
Phosphorus	15	22015	1.45	1.95	1.24	0.08	5.60
Sulfur	16	4429	0.32	0.43	0.26	0.04	11.83
Chlorine	17	51613	4.53	6.07	3.38	0.18	4.02
Silver	47	127532	21.98	29.46	5.38	0.76	3.45
**Sum**		**74.64**	**100.00**	**100.00**

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
