# Peer review of "Synthesis of Bioactive Silver Nanoparticles by a Pseudomonas Strain Associated with the Antarctic Psychrophilic Protozoon Euplotes focardii"

_marinedrugs, 2020, doi:10.3390/md18010038_

Round 1

Reviewer 1 Report

The article devoted to biosynthesis of silver nanoparticles (SNPs) with Pseudomonas strain. Also, authors report that obtained SNPs have higher antimicrobial activity comparison with SNPs obtained by chemical way. This effect is explained by coating SNPs with lipids or proteins during biosynthesis. And it seems that capped SNPs can be effective against pathogenic bacteria in spite of wide size distribution (which is often affect bad for antimicrobial properties).

 After reading the article I had a few questions\comments:

(line 76) …a rapid, simple, cost-effective approach.., but using endemic (line 79) is not associated with cheap and available method, 24 h for SNPs synthesis (line 89) are not like rapid too. (line 90) …22°C (the optimal growing temperature 89 of Pseudomonas sp. ef1)… but at line 82 authors claim that « …not surviving if exposed over 10 °C…» Could You explain it for me? Its known that AgNO3 is strong oxidizer and it’s hard to imagine how can alive organisms survive in so hard conditions. Could you explain if Ag ions reduce themselves due to oxidation and killing Pseudomonas sp? Is the strain alive after SNPs synthesis? Or bacteria reduce Ag in the process of life (by waste products, metabolites)? Figure 1 – All presented pictures have terrible It’s impossible to make any conclusions (size distribution) based on them. For example, on «B’» nobody can see scale. If it’s authors mistake or be incorrect loading of pictures by MDPI system  – it must be remake. The same problem with Fig. 2 and 3. (Line 149-150) The conclusion is not substantiated. (Figure 3) «B-» replace by «B)» (Figure 3) …asteriks… and Obeliks??)) replace by «asterisk» (Figure 3 B) XRD analysis show that there are a lot of AgO particles (amount is comparable with SNPs because of area of both peaks), also their size is bigger than SNPs (because of half width of peaks). So the presence of such a large number of AgO particles should affect the antibacterial properties of the system and should be mentioned in the introduction and their role is discussed in conclusion. (Figure 3 B) Using Scherrer equation let us determine average size of SNPs and compare it with TEM and SEM data. (Line229-232) I guess authors should explain, why they choose exactly such chemical method (sodium citrate) to obtain SNPs. There is no information about sizes and structure of obtained SNP. May be they have wide size distribution, agglomerates, ect. One may ask that if they specially obtained bad particles, in order to get good results when compared with them.

Author Response

Reviewer 1.

#1: The article devoted to the biosynthesis of silver nanoparticles (SNPs) with Pseudomonas strain. Also, authors report that obtained SNPs have higher antimicrobial activity comparison with SNPs obtained by chemical way. This effect is explained by coating SNPs with lipids or proteins during biosynthesis. And it seems that capped SNPs can be effective against pathogenic bacteria in spite of wide size distribution (which is often affect bad for antimicrobial properties).

ANSWER: Biologically synthesized nanoparticles coated with a lipid/protein layer confers physiological solubility and stability, which are essential to prevent aggregation as well as providing protection from temperature and light, and control drug release. The uptake of the SNP is important for antimicrobial activity. It is proposed that intimate contact between AgNPs and organisms may enhance the transfer of Ag ions to the bacterial cell, whilst bacterial degradation of the biological capping agent promotes the release of silver ions. Such interactions have been described as Trojan-horse mechanisms and improve AgNP efficacy.

#2: (line 76) …a rapid, simple, cost-effective approach.., but using endemic (line 79) is not associated with cheap and available method, 24 h for SNPs synthesis (line 89) are not like rapid too.

ANSWER: The term “endemic” is not referred to Pseudomonas sp. ef1 but to the host E. focardii (line 82). We removed from the text “rapid”.

#3: (line 90) …22°C (the optimal growing temperature 89 of Pseudomonas sp. ef1)… but at line 82 authors claim that « …not surviving if exposed over 10 °C…» Could You explain it for me?

ANSWER: « …not surviving if exposed over 10 °C…» is not referred to Pseudomonas sp. ef1 but to the host E. focardii (line 85). We better specify it in the text.

#3: Its known that AgNO3 is strong oxidizer and it’s hard to imagine how can alive organisms survive in so hard conditions. Could you explain if Ag ions reduce themselves due to oxidation and killing Pseudomonas sp? Is the strain alive after SNPs synthesis? Or bacteria reduce Ag in the process of life (by waste products, metabolites)?

ANSWER: Pseudomonas sp. ef1 metabolizes several pollutants, including diesel and heavy metal (Ramasamy Kesava, Ph.D. thesis). Pseudomonas sp. ef1 genome analysis (Ramasamy et al., 2019) also revealed nitrate reductase encoding genes that may help in tolerating AgNO3. We checked the toxicity of AgNO3 on Pseudomonas sp. ef1. It can stand up (survive up?) to 4 mM of AgNO3. We checked the viability of Pseudomonas sp. ef1 after SNP synthesis by plating on an agar medium a few drops of the solution and cells were still alive producing colonies.

#4: Figure 1 – All presented pictures have terrible It’s impossible to make any conclusions (size distribution) based on them. For example, on «B’» nobody can see scale. If it’s authors mistake or be incorrect loading of pictures by MDPI system  – it must be remake. The same problem with Fig. 2 and 3.

ANSWER: most probably it must have been an incorrect loading of pictures by the MDPI system because in the original version the scale is clearly visible. We also increased the size of fig. 2 and 3.

#5: (Line 149-150) The conclusion is not substantiated. (Figure 3) «B-» replace by «B)» (Figure 3) …asteriks… and Obeliks??)) replace by «asterisk» (Figure 3 B) XRD analysis show that there are a lot of AgO particles (amount is comparable with SNPs because of area of both peaks), also their size is bigger than SNPs (because of half width of peaks). So the presence of such a large number of AgO particles should affect the antibacterial properties of the system and should be mentioned in the introduction and their role is discussed in conclusion. (Figure 3 B) Using Scherrer equation let us determine average size of SNPs and compare it with TEM and SEM data.

ANSWER: (Line 149-150) The IR absorption reflects the biochemical composition of the sample. In particular, specific marker bands of proteins, lipids, and carbohydrates have been systematically described in the literature. We added the proper references in the revised text.  

We got four intense peaks at 38.95°, 45.12°, 65.39° and 78.12° which corresponds to Ag. According to the references reported below, the additional and yet unassigned peaks observed near to 27.47°, 32.11° and 46.21°,55.1 and 57.8 along with Bragg peaks representative of fcc silver nanocrystals, were assigned to the crystallization of bio-organic phase of compound or protein occurs on the surface of the silver nanoparticles. Similar results are reported by silver nanoparticles synthesized using Mangifera indica leaf extract, geranium leaves, mushroom extract, and Coleus aromaticus leaf extract. These peaks may also correspond to AgNO3 which has not been reduced and remained in the sample in minute quantity (since the XRD peaks are narrow). We added this interpretation in the revised text (lane 168).

References:

Brajesh Kumar, Kumari Smita ,Luis Cumbal,Alexis Debut. Green synthesis of silver nanoparticles using Andean blackberry fruit extract . Saudi Journal of Biological Sciences,24 (2017), pp.45-50. https://doi.org/10.1016/j.sjbs.2015.09.006

Mostafa M.H.Khalil,Eman H.Ismail,Khaled Z.El-Baghdady,DoaaMohamed. Green synthesis of silver nanoparticles using olive leaf extract and its antibacterial activity.Arabian Journal of Chemistry,7(2014), pp.  1131-1139 https://doi.org/10.1016/j.arabjc.2013.04.007

R. Sathyavathi, M.B.M. Krishna, S.V. Rao, R. Saritha, D.N. Rao.Biosynthesis of silver nanoparticles using coriandrum sativum leaf extract and their application in nonlinear optics.Adv. Sci. Lett., 3 (2010), pp. 1-6

D. Philip, “Mangifera indica leaf-assisted biosynthesis of well-dispersed silver nanoparticles,” Spectrochimica Acta Part A: Molecular and Biomolecular Spectroscopy, vol. 78, no. 1, pp. 327–331, 2011.

S. S. Shankar, A. Ahmad, and M. Sastry, “Geranium Leaf Assisted Biosynthesis of Silver Nanoparticles,” Biotechnology Progress, vol. 19, no. 6, pp. 1627–1631, 2003.

D. Philip, “Biosynthesis of Au, Ag and Au-Ag nanoparticles using edible mushroom extract,” Spectrochimica Acta. Part A, vol. 73, no. 2, pp. 374–381, 2009. 

The Full Width at Half Maximum (FWHM) values were calculated from the peaks to measure the size of the nanoparticles. The average size of synthesized silver nanoparticles was calculated using Scherrer’s equation where Scherrer’s constant  value = 0.94 was selected due to the cubic and crystalline nature of the nanoparticles. To estimate FWHM, each of the four peaks was fitted with a Gaussian function. The FWHM of the fitted Gaussian curve is taken as FWHM of the peak. This could be done in the software origin 6.1. The average crystalline size of the silver nanoparticles is found to be 20nm which is in good agreement with our TEM analysis with the size ranging from 12.5 to 100 nm

Furthermore, we corrected B- into B) and asteriks into asterisks.

#6: (Line229-232) I guess authors should explain, why they choose exactly such chemical method (sodium citrate) to obtain SNPs. There is no information about sizes and structure of obtained SNP. May be they have wide size distribution, agglomerates, ect. One may ask that if they specially obtained bad particles, in order to get good results when compared with them.

ANSWER: We chosen the chemical method with sodium citrate reported in Dey, A., Dasgupta, A., Kumar, V. et al. Int Nano Lett (2015) 5: 223. https://doi.org/10.1007/s40089-015-0159-2. We confirmed the chemical synthesis of Agnps by uv spec analysis and we compared the antimicrobial activity with that from obtained from Pseudomonas sp e1.

Reviewer 2 Report

The manuscript submitted by Sandra Pucciarelli and co-authors concerns important research problems such as green symthesis, nanotechnology and antimicrobial activity. More precisely, the authors have undertaken to develop a fast and simple synthesis of AgNP using a new strain of Pseudomonas. Moreover, they used several different methods to chracterize obtained materials, such as SEM, TEM , EDAX, FTIR and XRD. It contains several interesting results. I find the presented manuscript interesting and valuable. None the less, there are a few questions and suggestions that came to my mind while reading the manuscript.

Please present electronic absorption spectra of the obtained nanoparticles. Please correct Figure 2. It is difficult to read and interprate. Please separete part A and part 2. Table shoul dbe bigger, anothe rpart too. Please explain the differences in the size of the optainde nanoparticles.  Fig. 3 should also be improved, becasue it is too small. Please explain why the synthesis took place in Lauria-Bartani medium, not the agar. Please provide the full content of this medium. Is it possible that the components of the medium can lead to a reduction of silver? Additionally to Figure 4, there should appear another figure presented dose dependent antimicrobial activity.

Author Response

The manuscript submitted by Sandra Pucciarelli and co-authors concerns important research problems such as green symthesis, nanotechnology and antimicrobial activity. More precisely, the authors have undertaken to develop a fast and simple synthesis of AgNP using a new strain of Pseudomonas. Moreover, they used several different methods to chracterize obtained materials, such as SEM, TEM , EDAX, FTIR and XRD. It contains several interesting results. I find the presented manuscript interesting and valuable. None the less, there are a few questions and suggestions that came to my mind while reading the manuscript.

#1: Please present electronic absorption spectra of the obtained nanoparticles.

ANSWER: the absorption spectrum is reported in the supplementary material FigS1 C.

#2: Please correct Figure 2. It is difficult to read and interprate. Please separete part A and part 2. Table shoul dbe bigger, anothe rpart too. Please explain the differences in the size of the optainde nanoparticles.  Fig. 3 should also be improved, becasue it is too small. Please explain why the synthesis took place in Lauria-Bartani medium, not the agar. Please provide the full content of this medium. Is it possible that the components of the medium can lead to a reduction of silver? Additionally to Figure 4, there should appear another figure presented dose dependent antimicrobial activity.

ANSWER: The synthesis did not took place in Lauria-Bartani medium. The bacteria biomass was added to 1mM AgNO3 prepared in distilled water. Therefore, it is not possible that the components of the medium lead to a reduction of silver. The composition of the Lauria-Bartani medium was added in the new version of the paper (line 205).

In biological synthesis, it is not possible to get nanoparticles of similar sizes as in chemical and physical methods. There are several other important factors that affect the synthesis and size of nanoparticles, including pH of the solution, temperature, concentration of the extracts (biomass in our case), and reaction time.  Also the pressure applied to the reaction medium affects the shape and size of the synthesized nanoparticles. Furthermore, microorganisms produce distinct intracellular and extracellular enzymes that affect nanoparticle size. Therefore, we can measure the average size of the formed nanoparticles.

We did not perform dose dependent antimicrobial activity. In this paper we reported a preliminary antibacterial activity with the synthesized AgNPs. We are planning to perform a dose dependent antibacterial activity for a future work.

Finally, we also improved Fig 3.

Round 2

Reviewer 1 Report

Thanks author for detailed and clear answers. The article was finalized and became more correct.

Line 118-120 You are talking about SEM, but in new version of paper I don’t see SEM images at all. The authors loaded bigger version of figure 2 A,A’ (fig. 2 B,B’ were deleted as I understand), but its resolution is still inappropriate.

Line 166-171 Interpretation of XRD results:

The authors determine the unknown peaks referring to [53] paper (Sathyavathi, R., Krishna, M.B.M., Rao, S.V., Saritha, R., Rao, D.N. Biosynthesis of silver nanoparticles using Coriandrum Sativum leaf extract and their application in nonlinear optics. Adv.Sci. Lett., 2010 , 3, 1-6.) as «probably correspond to the crystallization of the bio-organic phase occurring on the AgNPs surface». It is wrong because:

The unknown peak’s position in this paper are approximately: 28⁰; 32,3⁰; 46,3⁰; 55⁰; 57,5⁰. The unknown peak’s position in [53] are: 38⁰ and 55⁰. – We can see just one overlap – it is not enough for making a conclusion that all of this peaks have common origin. Line 166-167, 174-175: angles «38,44,65,78 corresponding to plane values of (1 1 1), (2 0 0), (2 2 0) and (3 1 1)…» But in [53] authors write: «…peaks at 44.50, 52.20 and 76.7 corresponding

to the (111), (200) and (220) facets…» which is wrong. Authors of [53] made a simple mistake in interpretation of the XRD results. It can be a reason not to refer to XRD part of [53]

It’s possible to determine the source of unknown peaks using JCPDS/ICDD data bases or, for example: Hanawalt J.D., Rihh H.W., Frevel L.K. Chemical Analysis by X-Ray Diffraction. Classification and Use of X-Ray Diffraction Patterns // Industrial and engineering chemistry. – 1938. – Vol. 10. – № 7. – Р. 457–512. - Predecessor of JCPDS/ICDD data bases but in a free access (VERSION WITH TABLE ΧΙ INDEX TO POWDER DIFFRACTIOS DATA FOR 1000 CHEMICAL AL SUBSTASCES).

Then using data about interplanar distance ‘d’ of alleged substance (In your case it would be different silver compounds Ag2O, AgNO3..) and Wulff–Bragg's formula 2dsinθ=nλ, you can understand what substance was an origin of unknown peaks.

«…crystallization of the bio-organic phase occurring on the AgNPs surface…» - There is no reason for the crystallization of the bioorganic phase on the surface of silver nanoparticles.

Author Response

#1: Thanks author for detailed and clear answers. The article was finalized and became more correct.

Line 118-120 You are talking about SEM, but in new version of paper I don’t see SEM images at all. The authors loaded bigger version of figure 2 A,A’ (fig. 2 B,B’ were deleted as I understand), but its resolution is still inappropriate.

Answer

The SEM images were moved to the supplementary materials.

We improved the resolution of the TEM images

#2. Line 166-171 Interpretation of XRD results:

The authors determine the unknown peaks referring to [53] paper (Sathyavathi, R., Krishna, M.B.M., Rao, S.V., Saritha, R., Rao, D.N. Biosynthesis of silver nanoparticles using Coriandrum Sativum leaf extract and their application in nonlinear optics. Adv.Sci. Lett., 2010 , 3, 1-6.) as «probably correspond to the crystallization of the bio-organic phase occurring on the AgNPs surface». It is wrong because:

The unknown peak’s position in this paper are approximately: 28⁰; 32,3⁰; 46,3⁰; 55⁰; 57,5⁰. The unknown peak’s position in [53] are: 38⁰ and 55⁰. – We can see just one overlap – it is not enough for making a conclusion that all of this peaks have common origin. Line 166-167, 174-175: angles «38,44,65,78 corresponding to plane values of (1 1 1), (2 0 0), (2 2 0) and (3 1 1)…» But in [53] authors write: «…peaks at 44.50, 52.20 and 76.7 corresponding

to the (111), (200) and (220) facets…» which is wrong. Authors of [53] made a simple mistake in interpretation of the XRD results. It can be a reason not to refer to XRD part of [53]

It’s possible to determine the source of unknown peaks using JCPDS/ICDD data bases or, for example: Hanawalt J.D., Rihh H.W., Frevel L.K. Chemical Analysis by X-Ray Diffraction. Classification and Use of X-Ray Diffraction Patterns // Industrial and engineering chemistry. – 1938. – Vol. 10. – № 7. – Р. 457–512. - Predecessor of JCPDS/ICDD data bases but in a free access (VERSION WITH TABLE ΧΙ INDEX TO POWDER DIFFRACTIOS DATA FOR 1000 CHEMICAL AL SUBSTASCES).

Then using data about interplanar distance ‘d’ of alleged substance (In your case it would be different silver compounds Ag2O, AgNO3..) and Wulff–Bragg's formula 2dsinθ=nλ, you can understand what substance was an origin of unknown peaks.

«…crystallization of the bio-organic phase occurring on the AgNPs surface…» - There is no reason for the crystallization of the bioorganic phase on the surface of silver nanoparticles.

Answer:

The peaks from our analysis at 27.47° and 32.11° were also observed in:

1.Suresh Mickymaray. One-step Synthesis of Silver Nanoparticles Using Saudi Arabian Desert Seasonal Plant Sisymbrium irio and Antibacterial Activity Against Multidrug-Resistant Bacterial Strains. Biomolecules 2019, 9(11),662; https://doi.org/10.3390/biom9110662 (MDPI publications)

2.Mohammed Rafi Shaik et., al .Plant-Extract-Assisted Green Synthesis of Silver Nanoparticles Using Origanum vulgare L. Extract and Their Microbicidal Activities. Sustainability 2018, 10, 913; doi:10.3390/su10040913(MDPI publications)

3.Gitishree Das, Jayanta Kumar Patra, Trishna Debnath, Abuzar Ansari, Han Seung Shin.Investigation of antioxidant, antibacterial, antidiabetic, and cytotoxicity potential of silver nanoparticles synthesized using the outer peel extract of Ananas comosus (L.). PLoS One. 2019; 14(8): e0220950.doi: 10.1371/journal.pone.0220950.

4. Mahendran Vanaja and Annadurai Gurusamy. Coleus aromaticus leaf extract mediated synthesis of silvernanoparticles and its bactericidal activity. Appl Nanosci (2013) 3:217–223DOI 10.1007/s13204-012-0121-9,3.

5.DaizyPhilip.Biosynthesis of Au, Ag and Au–Ag nanoparticles using edible mushroom extract. Spectrochimica Acta Part A: Molecular and Biomolecular Spectroscopy,Volume 73, Issue 2, 15 July 2009, Pages 374-381)

6. Eman Zakaria Gomaa. Antimicrobial, antioxidant and antitumor activities of silver nanoparticles synthesized by Allium cepa extract: A green approach. J Genet Eng Biotechnol. 2017 Jun; 15(1): 49–57.doi: 10.1016/j.jgeb.2016.12.002

7. Nausheen Bibia, Qurban Alib , Zafar Iqbal Tanveerb , Hazir Rahmanc , and Muhammad Aneesa. Antibacterial efficacy of silver nanoparticles prepared using Fagoniacretica L. leaf extract, Inorganic and Nano-Metal Chemistry, DOI: 10.1080/24701556.2019.1661440

8.K. AnandalakshmiJ. Venugobal & V. Ramasamy . Characterization of silver nanoparticles by green synthesis method using Pedalium murex leaf extract and their antibacterial activity. Applied Nanoscience volume 6, pages399–408(2016).

The peak at 46.21° from our study was also observed in the following papers:

1. Shankar SSAhmad A and  Sastry M. Geranium Leaf Assisted Biosynthesis of Silver Nanoparticles. Biotechnol. Prog. 2003, 19, 1627−1631.

2. Brajesh Kumar, Kumari Smita ,Luis Cumbal,Alexis Debut. Green synthesis of silver

nanoparticles using Andean blackberry fruit extract . Saudi Journal of Biological

Sciences,24 (2017), pp.45-50. https://doi.org/10.1016/j.sjbs.2015.09.006

The peaks at 55.1° and 57.8° from our paper were also observed in the following papers:

1. S PonarulselvamC PanneerselvamK MuruganN AarthiK Kalimuthu, and S Thangamani. Synthesis of silver nanoparticles using leaves of Catharanthus roseus Linn. G. Don and their antiplasmodial activities. Asian Pac J Trop Biomed. 2012 Jul; 2(7): 574–580.doi: 10.1016/S2221-1691(12)60100-2.

In these papers, these unknown peaks were assigned to the crystallization of the bio-organic phase that might have occurred on the surface of the synthesized AgNPs. For this reason we propose that the unknown peaks from our study «probably correspond to the crystallization of the bio-organic phase occurring on the AgNPs surface».

All these references were added to the supplementary material.

Indeed our intention was not to compare our XRD results with those from the references since the synthesis was performed by different sources, i.e. bacteria vs plants. Furthermore, the focus of our study was to analyze the peaks corresponding to silver nanoparticles and just to provide a possible explanation of the additional unknown peaks.